# Use of a Longer Aglycon Moiety Bearing Sialyl α(2→3) Lactoside on the Glycopolymer for Lectin Evaluation

**DOI:** 10.3390/polym15040998

**Published:** 2023-02-17

**Authors:** Ryota Adachi, Takahiko Matsushita, Tetsuo Koyama, Ken Hatano, Koji Matsuoka

**Affiliations:** 1Area for Molecular Function, Division of Material Science, Graduate School of Science and Engineering, Saitama University, Sakura, Saitama 338-8570, Japan; 2Medical Innovation Research Unit (MiU), Advanced Institute of Innovative Technology (AIIT), Saitama University, Sakura, Saitama 338-8570, Japan; 3Health Sciences and Technology Research Area, Strategic Research Center, Saitama University, Sakura, Saitama 338-8570, Japan

**Keywords:** glycopolymers, radical polymerizations, carbohydrates, glycosides, oligosaccharides, glycoclusters, fluorescence spectroscopy, lectins, affinity constant

## Abstract

A polymerizable alcohol having 9 PEG repeats was prepared in order to mimic an oligosaccharide moiety. Sialyl α(2→3) lactose, which is known as a sugar moiety of GM3 ganglioside, was also prepared, and the polymerizable alcohol was condensed with the sialyl α(2→3) lactose derivative to afford the desired glycomonomer, which was further polymerized with or without acrylamide to give water-soluble glycopolymers. The glycopolymers had higher affinities than those of glycopolymers having sialyl lactose moieties with shorter aglycon moieties.

## 1. Introduction

Sialyl α(2→3) lactose [Neu5Ac α(2→3)Gal β(1→4)Glc; SLac] is known as a saccharide chain of GM3 ganglioside, and it exhibits a variety of biological activities [1]. Although the binding specificity of carbohydrate-binding proteins for SLac is high, the typical biological response of SLac to the carbohydrate-binding proteins is not good [10^−3^ M (mM) range] [2]. A remarkable enhancement of the low affinity of carbohydrates for lectins was accomplished by means of the sugar-clustering effect presented by Lee et al. [3,4]. Glycopolymers can be used as sugar-cluster substances [5,6], and various glycopolymers have been used as multivalent-type carbohydrate substrates for determining lectin—carbohydrate interactions [7,8]. Our ongoing synthetic studies using multivalent-type carbohydrate substances such as heterofunctional derivatives [9], glycodendrimers [10], and glycopolymers [11] showed effective enhancement of protein—carbohydrate interaction. Glycopolymers having SLac moieties as epitopes for the carbohydrate chain recognition protein on the mumps virus [12] were prepared, and they showed inhibitory potency against mumps viruses [13]. In addition to the mumps virus, avian influenza viruses also recognize the *N*-acetyl-neuraminic acid (Neu5Ac) residue in the SLac moiety [14]. Figure 1 shows the synthetic assembly of SLac moieties by means of polymer support, in which SLac having a triethylene glycol unit (PEG3) or a hexaethylene glycol unit (PEG6) was used as a linker between SLac and the polymer backbone displaying an appropriate length. In addition, acrylamide (**AAm**) was used as a controller adjusting the densities of inter SLac distances. Glycopolymers having SLac with PEG3 aglycon, as shown in Figure 1 were used as inhibitors for the mumps virus, and a glycopolymer having a low sugar density showed the highest inhibitory potency [13].

In this study, we planned to synthesize glycopolymers having a nonaethylene glycol unit (PEG9) moiety as a longer linker arm than that of previously synthesized glycopolymers, as shown in Figure 1. Figure 2 shows an outline of the synthesis of glycopolymers having a longer linker arm. Biological evaluation of the glycopolymers for a lectin on the basis of fluorescence changes of the lectin was also carried out.

## 2. Materials and Methods

### 2.1. Materials

Unless otherwise stated, all commercially available solvents and reagents were used without further purification. Methanol (MeOH) and dichloromethane (DCM) were stored over 3 Å MS, and 1,2-dichloromethane was stored over 4 Å MS before use. Acrylamide was recrystallized from chloroform (CHCl_3_) before use. 17-Azido-3,6,9,12,15-pentaoxaheptadecan-1-ol (**1**) was prepared according to the previous reported method [15]. An imidate **6** used as a glycosyl donor was obtained by means of a known procedure [13]. Optical rotations were determined with a JASCO DIP-1000 digital polarimeter (JASCO Corporation, Tokyo, Japan). IR spectra were obtained using a Shimadzu IR Prestige-21 spectrometer (SHIMADZU CORPORATION, Kyoto, Japan). ^1^H NMR spectra were recorded at 400 MHz for ^1^H and at 101 MHz for ^13^C with a Bruker DPX-400 spectrometer (Bruker BioSpin MRI GmbH, Ettlingen, Germany) or at 500 MHz for ^1^H and at 126 MHz for ^13^C with a Bruker AVANCE 500 spectrometer (Bruker BioSpin MRI GmbH, Ettlingen, Germany) in chloroform-*d* (CDCl_3_), dimethyl sulfoxide-*d*_6_ (DMSO-*d*_6_), or deuterium oxide (D_2_O). Chemical shifts are expressed as parts per million (ppm, δ) and are relative to an internal tetramethylsilane (TMS) in CDCl_3_ (δ 0.0), CH_3_ in DMSO-*d*_6_ (δ 2.50), or HDO in D_2_O (δ 4.78) for ^1^H and CDCl_3_ in CDCl_3_ (δ 77.0), CH_3_ in DMSO-*d*_6_ (δ 39.51), or CH_3_ in MeOD (δ 49.0) for ^13^C. Ring-proton assignments in the ^1^H NMR spectra were made by a first-order analysis of the spectra, and these are supported by the results of homonuclear decoupling experiments and H–H or HMQC experiments. Elemental analyses were performed with a Fisons EA1108 (Fisons Co., Milan, Italy) on samples extensively dried at 50–60 °C over phosphorus pentoxide for 4–5 h. The reactions were monitored by thin layer chromatography (TLC) on a precoated plate of Silica Gel 60F_254_ (layer thickness, 0.25 mm; E. Merck, Darmstadt, Germany). For the detection of the intermediates, TLC sheets were dipped in (a) a solution of 85:10:5 (*v*/*v*/*v*) MeOH–*p*-anisaldehyde–concd H_2_SO_4_ and heated for a few minutes (for carbohydrate) or (b) an aq solution of 5 wt% KMnO_4_ and heated similarly (for the detection of C=C double bonds). Column chromatography was performed on silica gel (Silica Gel 60; 63–200 μm, E. Merck). Flash column chromatography was performed on silica gel (Silica Gel 60, spherical neutral; 40–100 μm, E. Merck). All the extractions were concentrated below 45 °C under diminished pressure. The weight-average molecular weights (s) were estimated by size-exclusion chromatography in 0.3 M aq NaCl using tandem-bonded Shodex SB-803 HQ and SB-804 HQ columns (SHOWA DENKO K.K., Tokyo, Japan). Calibration curves were obtained using pullulan standards (5.9, 11.8, 22.8, 47.3, 112, 212, 404, and 788 kDa; Shodex P-82). Wheat germ agglutinin (WGA; a lectin from Triticum vulgaris) was purchased from J-Oil Mils (J-OIL MILLS, INC., Tokyo, Japan) (Lot # 62015).

### 2.2. Synthesis

#### 2.2.1. 17-Azido-3,6,9,12,15-pentaoxaheptadecyl methanesulfonate (**2**)

To an ice-cold solution of azidoalcohol **1** (5.01 g, 16.3 mmol) and Et_3_N (2.70 mL, 19.5 mmol) in DCM (50 mL) was dropwise added methanesulfonyl chloride (1.50 mL, 19.4 mmol) at 0 °C under an Ar atmosphere. After stirring at 0 °C for 1 h, TLC showed a complete conversion of the starting alcohol, and the mixture was filtered through a pad of Celite. The filtrate was concentrated in vacuo and diluted with THF. The organic solution was successively washed with water and brine, dried over anhyd Na_2_SO_4_, filtered, and evaporated at diminished pressure to afford pure yellowish mesylate (6.25 g, 99%) as a syrup; *R*_f_ 0.53 [EtOAc]; ^1^H NMR (400 MHz, DMSO-*d*_6_) δ 4.32–4.26 (m, 2 H, OCH_2_), 3.69–3.63 (m, 2 H, OCH_2_), 3.62–3.49 (m, 20 H, OCH_2_ x10), 3.16 (s, 3 H, CH_3_); ^13^C NMR (101 MHz, DMSO-*d*_6_) δ 69.86 (OCH_2_), 69.82 (OCH_2_), 69.78 (OCH_2_), 69.74 (OCH_2_), 69.71 (OCH_2_), 69.28 (OCH_2_), 68.32 (OCH_2_), 36.87 (CH_3_); IR (NEAT) 2872 (ν_C-H_), 2110 (ν_N=N=N_), 1350 (ν_S=O_), 1174 (ν_S=O_), 1113 (ν_C-O_) cm^−1^.

#### 2.2.2. 26-Azido-3,6,9,12,15,18,21,24-octaoxahexacosan-1-ol (**3**)

A solution of KO*t*-Bt (5.26 g, 46.9 mmol) and triethylene glycol (11.0 mL, 82.6 mmol) in *t*-BuOH (60 mL) was refluxed for 1 h under an Ar atmosphere. Mesylate **2** (6.02 g, 15.6 mmol) was added to the mixture, and the mixture was further stirred for 2.5 h at refluxed temperature under an Ar atmosphere. TLC indicated the consumption of the starting materials and Celite filtration was performed for the reaction mixture. The filtrate was evaporated in vacuo; this was followed by chromatographic purification on silica gel with 20:1 (*v*/*v*) CHCl_3_—MeOH as the eluent to yield the desired azide alcohol **3** (6.03 g, 88%) as yellowish liquid; *R*_f_ 0.43 [4:1 EtOAc—MeOH (*v*/*v*)]; ^1^H NMR (400 MHz, DMSO-*d*_6_) δ 4.54 (t, 1 H, *J* = 5.5 Hz, OH), 3.60 (dd, 2 H, *J* = 5.6 & 4.4 Hz, OCH_2_), 3.58–3.45 (m, 30 H, OCH_2_), 3.45–3.36 (m, 4 H, OCH_2_); ^13^C NMR (101 MHz, DMSO-*d*_6_) δ 72.32 (OCH_2_), 69.80 (OCH_2_), 69.76 (OCH_2_), 69.67 (OCH_2_), 69.22 (OCH_2_), 60.19 (HOCH_2_), 49.98 (OCH_2_); IR (NEAT) 3482 (ν_O-H_), 2870 (ν_C-H_), 2110 (ν_N=N=N_), 1121 (ν_C-O_) cm^−1^.

#### 2.2.3. 26-Amino-3,6,9,12,15,18,21,24-octaoxahexacosan-1-ol (**4**)

A solution of azide alcohol **3** (5.33 g, 12.1 mmol) in MeOH (50 mL) was stirred for a few minutes at RT under an Ar atmosphere. Pd(OH)_2_/C (1.64 g) was added to the solution under an Ar atmosphere. The Ar gas was immediately replaced by H_2_ gas, and the suspension was stirred for 6.5 h at RT under a H_2_ atmosphere. TLC showed disappearance of the azide **3**, and the H_2_ gas was changed to Ar gas. The suspension was filtered through a pad of Celite, followed by concentration at diminished pressure to give the corresponding amine **4** (3.93 g) in 78% yield; *R*_f_ 0.08 [4:1 EtOAc—MeOH (*v*/*v*)].

#### 2.2.4. N-(26-Hydroxy-3,6,9,12,15,18,21,24-octaoxahexacosyl)acrylamide (**5**)

Acryloyl chloride (0.40 mL, 4.92 mmol) was dropwise added to a solution of amine **4** (1.01 g, 2.45 mmol), hydroquinone (6 mg, 50 μmol), and diisopropyl ethylamine (1.3 mL, 7.46 mmol) in MeOH (10 mL) at ice-cold temperature under an Ar atmosphere. The reaction mixture was stirred for 1.5 h at the same temperature and for 1 h at RT. The reaction mixture was concentrated in vacuo, and silica gel chromatography with 4:1 (*v*/*v*) EtOAc—MeOH of the resulting syrup afforded desired acryl amide **5** (0.350 g, 31%) as a yellow syrup; *R*_f_ 0.39 [2:1 EtOAc—MeOH (*v*/*v*)]; ^1^H NMR (500 MHz, DMSO-*d*_6_) δ 8.15 (br, 1 H, NH), 6.24 (dd, 1 H, *J*_trans_ = 17.1 and *J*_cis_ = 10.2 Hz, -CH=), 6.07 [dd, 1 H, *J*_gem_ = 2.2 Hz, =CH_2_ (trans)], 5.57 [dd, 1 H, =CH_2_ (cis)], 4.57 (br, 1 H, OH), 3.51 (m, 32 H, OCH_2_), 3.44 (t, 2 H, *J* = 5.8 Hz), 3.41 (dd, 4 H, *J* = 5.6 and 4.6 Hz), 3.28 (q, 2 H, *J* = 5.8 Hz); ^13^C NMR (126 MHz, DMSO-*d*_6_) δ 164.64 (C=O), 131.73 (-CH=), 125.01 (=CH_2_), 72.34 (OCH_2_), 69.81 (OCH_2_), 69.78 (OCH_2_), 69.71 (OCH_2_), 69.59 (OCH_2_), 69.04 (OCH_2_), 60.21 (OCH_2_).

#### 2.2.5. 28-Oxo-3,6,9,12,15,18,21,24-octaoxa-27-azatriacont-29-en-1-yl [Methyl(5-acetamido-4,7,8,9-tetra-O-acetyl-3,5-dideoxy-$\underset{̿}{D}$-glycero-α-$\underset{̿}{D}$-galacto-2-nonulopyranosyl)onate]-(2→3)-O-(2,4,6-tri-O-acetyl-β-$\underset{̿}{D}$-galactopyranosyl)-(1→4)-2,3,6-tri-O-acetyl-β-$\underset{̿}{D}$-glucopyranoside (**7**)

Powdered molecular sieves 4 A (2.26 g) was added to a solution of imidate **6** (2.16 g, 1.79 mmol) and acrylamide alcohol **5** (1.02 g, 2.29 mmol) in DCM (20 mL) under an Ar atmosphere, and the suspension was stirred at RT for 30 min under an Ar atmosphere. To the suspension was dropwise added BF_3_•OEt (0.45 mL, 3.58 mmol) at −15 °C under an Ar atmosphere and the mixture was stirred at the same temperature for 30 min, followed by further stirring at RT for 1 h. When TLC showed consumption of the starting materials, the suspension was passed through a pad of Celite. The filtrate was diluted with CHCl_3_, and the organic solution was successively washed with satd aq NaHCO_3_ and brine, dried over anhyd Na_2_SO_4_, filtered, and evaporated in vacuo. The residue yielded the desired glycoside **7** (0.953 g, 35%) and the corresponding hemiacetal **8** (1.13 g, 59%) after chromatographic purification on silica gel with 5:1 (*v*/*v*) EtOAc—MeOH as the eluent; *R*_f_ 0.33 [4:1 EtOAc—MeOH (*v*/*v*)]; ^1^H NMR (400 MHz, CDCl_3_) δ 6.78 (br s, 1 H, NHCO), 6.29 [dd, 1 H, *J*_trans_ = 17.0 Hz & *J*_gem_ = 1.8 Hz, CH_2_ (trans)], 6.17 (dd, 1 H, *J*_cis_ = 10.1 Hz, CH=), 5.61 [dd, 1 H, =CH_2_ (cis)], 5.54 (ddd, 1 H, *J*_7″,8″_ = 9.2 Hz, *J*_8″,9″b_ = 5.1 Hz, & *J*_8″,9″a_ =2.7 Hz, H-8″), 5.39 (dd, 1 H, *J*_6″,7″_ = 2.8 Hz, H-7″), 5.18 (t, 1 H, *J*_2,3_ = *J*_3,4_ = 9.3 Hz, H-3), 4.97–4.83 (m, 4 H, H-2, H-2′, H-4′, & H-4″), 4.67 (d, 1 H, *J*_1′,2′_ = 8.0 Hz, H-1′), 4.55 (d, 1 H, *J*_1,2_ = 8.0 Hz, H-1), 4.52 (dd, 1 H, *J*_2′,3′_ = 10.2 & *J*_3′,4′_ = 3.3 Hz, H-3′), 4.48–4.38 (m, 2 H, H-6 & H-9a″), 4.27–3.86 (m, 8 H), 3.84 (s, 3 H, COOMe), 3.76 –3.53 (m, 41 H, OCH_2_CH_2_), 2.58 (dd, 1 H, *J*_3″ax,3″eq_ = 12.7 Hz & *J*_3″eq,4″_ = 4.6 Hz, H-3″eq), 2.16, 2.09, 2.09, 2.08, 2.07, 2.06, 2.04, 2.03, 2.00, 1.85 (each s, 10 Ac), 1.67 (t, 1 H, *J*_3″ax,4″_ = 12.4 Hz, H-3″ax); ^13^C NMR (101 MHz, CDCl_3_) δ 170.99 (C=O), 170.77 (C=O), 170.62 (C=O), 170.51 (C=O), 170.44 (C=O), 170.37 (C=O), 169.90 (C=O), 169.77 (C=O), 169.70 (C=O), 168.09 (C=O), 165.79 (C=O), 131.25 (CH=), 126.17 (CH_2_=), 101.13 (C-1′), 100.78 (C-1), 96.92 (C-2″), 77.48, 77.16, 76.84, 76.41, 73.51, 72.83, 72.15, 71.91, 71.52, 70.81, 70.71, 70.68, 70.65 (OCH_2_), 70.37, 70.32, 70.06, 69.97, 69.46, 69.21, 67.91, 67.45, 67.05, 62.44, 62.36 (C-6), 61.66 (C-6′), 60.52, 53.26 (OMe), 49.26 (C-5″), 39.45 (OCH_2_), 37.54 (C-3″), 23.30 (CH_3_), 21.65 (CH_3_), 21.08 (CH_3_), 20.96 (CH_3_), 20.92 (CH_3_), 20.86 (CH_3_), 20.82 (CH_3_), 20.74 (CH_3_); IR (KBr) 3431 (ν_N-H_), 3277 (ν_N-H_), 2943 (ν_C-H_), 2880 (ν_C-H_), 1748 (ν_C=O_, ester), 1665 (ν_C=O_, amide I), 1545 (δ_N-H_, amide II), 1371 (δ_C-O_), 1043 (ν_C-O-C_) cm^−1^; [α]_D_^25^ = −4.96° (*c* 1.16, CHCl_3_); MALDI-TOF MS calcd for [M+Na]^+^: 1539.584. Found: *m*/*z* 1539.488, calcd for [M+K]^+^: 1555.558. Found: *m*/*z* 1555.476.

Anal. Calcd for C_65_H_100_N_2_O_38_•3 H_2_O: C, 49.68; H, 6.80; N, 1.78. Found: C, 49.64; H, 6.69; N, 1.57.

#### 2.2.6. 28-Oxo-3,6,9,12,15,18,21,24-octaoxa-27-azatriacont-29-en-1-yl (5-Acetamido-3,5-dideoxy-$\underset{̿}{D}$-glycero-α-$\underset{̿}{D}$-galacto-2-nonulopyranosyl)-(2→3)-O-(β-$\underset{̿}{D}$-galactopyranosyl)-(1→4)-β-$\underset{̿}{D}$-glucopyranoside (**9**)

Acetate **7** (0.902 g, 0.594 mmol) was treated with 0.1 M methanolic NaOMe (9 mL) at RT for 4 h under an Ar atmosphere. TLC indicated complete conversion of **7**, and Dowex 50W-x8 (H^+^) was added to the reaction mixture until pH reached pH 7 on pH paper. The suspension was passed through a pad of cotton, and the filtrate was concentrated in vacuo. The residue was allowed to react with 0.05 M aq NaOH (9 mL) at RT for 3 h under an Ar atmosphere. To the reaction mixture was added Dowex 50W-x8 (H^+^) until pH reached pH 7 on pH paper and the mixture was filtered. The filtrate was directly lyophilized and gave a white powdery mass **9** (0.586 g) in 91% yield: *R*_f_ 0.56 [5:4:1 CHCl_3_—MeOH—H_2_O (*v*/*v*)]; ^1^H NMR (400 MHz, D_2_O) δ 6.33 (dd, 1 H, *J*_trans_ = 17.2 Hz & *J*_cis_ = 10.0 Hz, CH=), 6.24 [dd, 1 H, *J*_gem_ = 1.6 Hz, =CH_2_ (trans)], 5.81 [dd, 1 H, =CH_2_ (cis)], 4.57 (d, 1 H, *J*_1′2′_ = 8.0 Hz, H-1′), 4.55 (d, 1 H, *J*_1,2_ = 8.0 Hz, H-1), 3.52 (t, 2 H, *J* = 5.2 Hz, CH_2_N), 3.38 (t, 1 H, *J*_2,3_ = 8.3 Hz, H-2), 2.79 (dd, 1 H, *J*_3″ax,3″eq_ = 12.6 Hz & *J*_3″eq,4″_ = 4.6 Hz, H-3″eq), 2.07 (s, 3 H, NAc), 1.89 (t, 1 H, *J*_3″ax,4″_ = 12.2 Hz, H-3″ax); ^13^C NMR (101 MHz, D_2_O) δ 175.02 (C-1″), 172.77 [C=O(Ac)], 168.63 [C=O(acryloyl)], 129.91 (CH=), 127.40 (=CH_2_), 102.66 (C-1′), 102.15 (C-1), 99.24 (C″-2), 78.30, 75.50 (C-3′), 75.09, 74.78 (C-2′), 74.30, 73.05, 72.83 (C-2), 71.38, 69.59, 69.45, 68.74, 68.15, 67.95, 67.56, 62.79, 60.95, 60.09, 51.68 (C-5″), 39.28 (C-3″), 39.03 (CH_2_N), 22.07 (CH_3_); IR (KBr) 3399 (ν_O-H_, ν_N-H_), 2926 (ν_C-H_), 2887 (ν_C-H_), 1647 (ν_C=O_ amide I), 1558 (δ_N-H_), 1070 (ν_C-O-C_)cm^−1^; [α]_D_^25^ = −2.11° (*c* 1.13, H_2_O); MALDI-TOF MS calcd for [M+Na]^+^: 1105.463. Found: *m*/*z* 1105.656.

Anal. Calcd for C_44_H_78_N_2_O_28_•4 H_2_O: C, 45.75; H, 7.50; N, 2.43. Found: C, 45.91; H, 7.54; N, 2.21. 

#### 2.2.7. Radical Polymerization

A solution of appropriate amounts of carbohydrate monomer and acrylamide (**AAm**) in 1:1 (*v*/*v*) deionized water—DMSO was deaerated under reduced pressure for a few minutes, and then *N*,*N*,*N*′,*N*′-tetramethylethylenediamine (TEMED) (0.2 molar equivalent for the corresponding carbohydrate monomer) and ammonium persulfate (APS) (0.1 molar equivalent for the corresponding carbohydrate monomer) were added under an Ar atmosphere at RT. The mixture was stirred at RT for an appropriate time and diluted with 0.1 M aq pyridine—acetic acid buffer (pH 5.2). The viscous solution was dialyzed against distilled water, followed by lyophilization to provide the corresponding white powdery glycopolymers **10a**~**10d**. The results of radical polymerization are summarized in Table 1.

## 3. Results and Discussion

In our ongoing synthetic studies, glycomonomers having a shorter polymerizable linker than that in this study have been reported [16]. The radical polymerization reaction proceeded smoothly to afford the corresponding glycopolymers having various SLac-saccharide densities, and biological evaluations of the various glycopolymers for wheat germ agglutinin (WGA) were carried out. The synthetic assembly of SLac having a longer polymerizable linker is described.

### 3.1. Monomer Synthesis

#### 3.1.1. Synthesis of PEG9 Linker

Our synthetic target as a polymerizable alcohol **5** is illustrated in Figure 1. The elongation of oxyethylene repeating units was achieved by means of simple chemical reactions. Thus, a known azide alcohol **1** [16] was converted to the corresponding mesylate **2** in quantitative yield. An elongation of the triethylene glycol unit for **2** was carried out by using KO*t*-Bu as a nucleophilic reagent in order to activate the alcohol moiety generating the alkoxide in situ. The replacement of mesylate by triethylene glycol proceeded smoothly at refluxed temperature in *t*-BuOH to afford the corresponding azide alcohol **3** having nine polyethylene repeating units (PEG9) in 88% yield. The reduction of azide was achieved by means of typical hydrogenolysis conditions under a H_2_ atmosphere giving the corresponding amine **4** in 78% yield. 

As the aminoalcohol **9** having a PEG9 linker had been obtained, our attention turned to the acryloylation of the amine of **9**. Consequently, acryloyl chloride was added to the solution of amine **4** in the presence of Et_3_N in MeOH; however, none of the desired **5** was obtained, as shown in entry 1 in Table 2. The amide formation reaction proceeded on TLC, and subsequent polymerization of the acrylamide functional group would have occurred. In addition, a large amount of triethylamine-hydrochloride salt produced in the reaction hindered the isolation of pure **5** from the reaction mixture. The use of an ion exchange resin instead of Et_3_N was suggested as a means of avoiding this situation. Therefore, IRA400J was used as the anion exchange resin for the reaction; however, the resin did not function as an acid scavenger, and no product at all was obtained, as shown in entry 2 in Table 2. Although amide formation would have proceeded in the reaction conditions, the polymerization of the acrylamide moiety also occurred simultaneously. It was found that protection of the free radical in the reaction mixture was needed to avoid this side reaction [17]. A free radical inhibitor was used for the preparation of acrylamide alcohol **5** [18]. Thus, a small amount of hydroquinone (HQ) was applied to the acryloylation, as shown in entry 3 in Table 2. The reaction proceeded, and the desired acrylamide derivative was obtained in 24% yield. We tried changing the base in addition to controlling the amounts of the base and the HQ (entry 3~6). When 3 molar equivalents of *N*,*N*-diisopropylethylamine (DIPEA) and 0.02 molar equivalent of HQ were used for the reaction, the best yield of acrylamide **5** was obtained. 

#### 3.1.2. Synthesis of SLac-PEG9 Monomer

Given the success in the preparation of acrylamide alcohol, our next area of interest was the incorporation of the alcohol into SLac. Because of the efficiency and easy handling associated with the glycosidation reaction between a carbohydrate donor and an acceptor, the use of the imidate strategy presented by Schmidt [19] was selected in this case (Figure 2). The trisaccharide moiety of SLac was prepared using the method previously reported [20,21]. Thus, a glycosidation reaction between a freshly prepared imidate **6** as a glycosyl donor and acrylamide alcohol **5** as a glycosyl acceptor was carried out in the presence of BF_3_•OEt as a Lewis acid catalyst to afford the desired SLac derivative **7** in 35% yield. Since we were not able to remove moisture from the reaction mixture prior to the condensation reaction, the use of two molar equivalents of BF_3_•OEt was needed. A previous report [22] indicated that PEG typically binds 2–3 water molecules per ethylene oxide unit, and a low yield of the glycosidation was therefore encountered. In addition to the glycoside **7**, the corresponding hemiacetal **8** was also obtained in 59% yield. The hemiacetal **8** was produced by the attack of water molecules on the oxocarbenium ion as an intermediate. Zemplén’s transesterification [23] followed by subsequent saponification for fully protected SLac glycoside **7** furnished the desired glycomonomer **9** in 95% yield in two steps. 

#### 3.1.3. Polymerization of SLac-PEG9 Monomer

Working towards the completion of the preparation of SLac having polymerizable PEG9 aglycon, our next objective was the synthetic assembly of the SLac moieties by means of radical polymerization. We polymerized various glycomonomers in aqueous media using a convenient radical polymerization technique [11,24,25]. Thus, the glycomonomer **9** was allowed to polymerize with or without acryl amide as a mediator for controlling the distance between SLac residues in a 1:1 water—DMSO solvent system in the presence of APS and TEMED as shown in Figure 3. The reaction proceeded smoothly to yield the corresponding water-soluble glycopolymers **10a**~**10d**, and the results are summarized in Table 1. It is interesting to note that, according to the results of the polymerization, when a higher ratio of **AAm** was used as comonomer in the polymerization reaction, the yields were worse. It was found that the glycomonomer **9** having a PEG9 moiety as the aglycon showed higher polymerization activity than that of **AAm**. We concluded that the reason for the effect of the PEG9 moiety on the yields was that PEG9 moiety displayed higher hydrophilic activity in this case. The polymer compositions for each polymerization were close to the charged ratio of the corresponding monomers. 

^1^H NMR spectra of the monomer **9**, homopolymer **10a**, and copolymer **10b** as an example of the copolymers are shown in Figure 3. Acrylamide moieties at approximately 5.8~6.5 ppm in monomer **9** in Figure 3a completely disappeared after polymerization, as shown in Figure 3b,c). These spectra supported the complete incorporation of the SLac moiety into the polymer. Characteristic protons due to the sialic acid moiety at approximately 2.8 ppm for H-3″_eq_ and at approximately 1.9 ppm for H-3″_ax_ were clearly observed in each spectrum. In addition, each anomeric proton signal of galactose and glucose residues of SLac trisaccharide appeared as two doublets at approximately 4.5 ppm. 

#### 3.1.4. Biological Response to WGA Determined by a Fluorometric Assay

We examined the biological activities of the series of glycopolymers that were prepared for carbohydrate-binding proteins prior to application as inhibitors for various pathogenic viruses. We recently carried out biological evaluation of lectins as carbohydrate-binding proteins [11], and the same procedure was used in this study. Wheat germ agglutinin (WGA) [26] is one of the well-known lectins that recognizes mainly *N*-acetyl-D-glucosamine and their oligomers [27,28]; however, *N*-acetyl-neuraminic acid (Neu5Ac) is also recognized [29,30]. In our previous studies, investigation with a series of glycopolymers as carbohydrate substrates was carried out by means of changes in fluorescence emission from tryptophane (Trp) residues of WGA on its own when aliquots of the glycopolymer solution were added to the analytical solution. Fluorometric analyses of a glycopolymer **10b** as an example in 0.65 μM of WGA solution in 50 mM Tris-HCl buffer (pH 7.5) at 4 °C is shown in Figure 4. The WGA solution was maintained at 4 °C ± 0.1 °C for all measurements in order to avoid nonspecific binding of the glycopolymer—WGA, and specific excitation of Trp on WGA was performed at λ_295_ nm [31,32]. The fluorescence spectrum of the WGA solution upon addition of an aliquot of a polymer solution **10b** was monitored, and the spectra are shown in Figure 4a. The intensity of the spectrum gradually increased, and each value at 348 nm of the spectra was recorded. Figure 4b shows changes in the intensities of the emission spectra. A Hill plot analysis [33] was carried out by using the parameters shown as Figure 4c, and the results provided the corresponding association constant *K*_a_ of 11.3 × 10^5^ M^−1^, which is 35 times higher than that of the monomeric SLac derivative **9**. Other glycopolymers that were prepared were analyzed in the same manner as that described for **10b**, and all of the profiles of the fluorescence spectra of the glycopolymer—WGA interaction are shown in the Appendix A. Table 3 summarizes the results of biological evaluations based on fluorometric assays. Changes in fluorescence intensities of the glycopolymers as represented by Δ*F*′/*F*_0_ were 17% to 75%, and the *K*_a_ values were 0.32 × 10^5^ M^−1^ to 11.3 × 10^5^ M^−1^. The *K*_a_ values of the glycopolymers were higher than the *K*_a_ value of a monomeric compound **9**, and this phenomenon strongly supported a positive sugar-clustering effect [3]. Glycopolymer **10b** used as the substrate showed the highest *K*_a_ value. In addition, Gibbs free energies Δ*G*^0^_a_ were calculated from the relationship Δ*G*^0^_a_ = −RTln*K*_a_, and the values of −Δ*G*^0^_a_ of glycopolymers were 24~32 kJ/mol on the basis of the sugar unit.

In the current study, a series of SLac polymers having shorter aglycon moieties was prepared, and preliminary biological evaluations for WGA using fluorometric assays were carried out. The results suggested that there is a relationship between a larger degree of freedom of the sugar moiety in SLac polymers having longer aglycon than that of our current study [16] and the sugar densities on the corresponding glycopolymers. Further chemical manipulations and control of the sugar densities of the SLac derivatives are now being carried out, and the results will be reported elsewhere in the near future.

## 4. Conclusions

Simple hydrophilic mimetics for the oligosaccharide moiety portion were achieved by means of PEG repeating units, and a polymerizable alcohol having 9 PEG repeats was prepared. Sialyl α(2→3) lactose, which is known as a sugar moiety of GM3 ganglioside, was also synthesized and was coupled with the polymerizable alcohol to afford the desired glycomonomer, which was further polymerized with or without acrylamide to give the corresponding water-soluble glycopolymers. The glycopolymers had higher affinities than those of glycopolymers having sialyl lactose moieties with shorter aglycon moieties.

## Data Availability

Data are contained within the article or Appendix A.

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
