# Peer review of "Use of a Longer Aglycon Moiety Bearing Sialyl α(2→3) Lactoside on the Glycopolymer for Lectin Evaluation"

_polymers, 2023, doi:10.3390/polym15040998_

Round 1

Reviewer 1 Report

The article which is submitted for publication to polymers is entitled ; «  Use of a Longer Aglycon Moiety Bearing Sialyl α(2→3) Lactoside on the Glycopolymer for Lectin Evaluation ». It is co-authored by R. Adachi, T. Matsushita, T. Koyama, K. Hatano &  K. Matsuoka

The work described in the manuscript deals with an essential aspect of the field of protein-carbohydrate interactions. It provides a convincing synthetic preparation of a glycopolymer, where the biologically important trisaccharide  Neu5Ac α(2→3)Gal β(1→4)Glc is linked to a nonaethylene glycol aglycon. The synthetic route is well described, as is the structural characterization of the obtained glycopolymer. The authors complete their work by studying the binding of the glycopolymer to a lectin (What Germ Agglutinin) that primarily binds to N-acetyl-D-glucosamine (GlcNAc) and is also reported to interact with sialic-acid-containing glycoconjugates and oligosaccharides. They observed a  noticeable increase in the affinity of the lectin towards the synthesised glycopolymer (up to 35 times more)

The presentation of the manuscript is clear, and all synthetic steps are well described. The relevant references are correctly cited.

Before publishing, the authors should care for the following minor points.

1.      In Figure 1. I presume « specer » should be replaced with « spacer. »

2.      The authors use different names to describe the same chemical motif: Sialic acid, N-acetyl-neuraminic acid (NANA)

3.      The format of the following references, 2, 4, 14, 17, 19, 20, 24, 30, should be corrected, and capital letters not used.

Author Response

To Reviewer #1

     Thank you very much for the valuable comments and suggestions. 

Q1.     In Figure 1. I presume « specer » should be replaced with « spacer. »

A1.     Thank you very much for your findings.  It was typo, and the spell has been replaced.  The Figure 1 was replaced.

Q2.     The authors use different names to describe the same chemical motif: Sialic acid,

N-acetyl-neuraminic acid (NANA)

A2.     NANA at line #312 was changed to Neu5Ac.

Q3.     The format of the following references, 2, 4, 14, 17, 19, 20, 24, 30, should be

corrected, and capital letters not used.

A3.     All references mentioned Q3 are revised according to the referee’ comments.

Reviewer 2 Report

In this manuscript, the Prof. Koji Matsuoka and coworkers reported “ Use of a Longer Aglycon Moiety Bearing Sialyl α(2→3) Lacto-2 side on the Glycopolymer for Lectin Evaluation” . In this paper, A polymerizable alcohol having 9 PEG repeats was prepared to mimic an oligo-13 saccharide moiety. Sialyl α(23) lactose was also prepared and the polymerizable alcohol was condensed with the sialyl α(23) lactose  derivative to afford the desired glycomonomer, The glycopolymers had higher affinities than those of glycopolymers having sialyl lactose moieties with shorter aglycon moieties.The manuscript is novel, results are impressive, and the manuscript can be considered for publication; however, there are some minor comments: 

1.    The 1H spectra of compound 2, there is a singlet peak at 2.5, what’s this?2.    It seems that compound 4 purity is not very good, please purified and rerun the proton and carbon. 

3.  The compound 5 integral is not accurate with your paper, page 4, line 133.

Author Response

To Reviewer #2

     Thank you very much for the valuable comments and suggestions. 

Q1.     The 1H spectra of compound 2, there is a singlet peak at 2.5, what’s this?

A1.     Thank you very much for your comments.  The signal at 2.5 ppm is from DMSO of DMSO-d6 used as solvent for NMR.

Q2.     It seems that compound 4 purity is not very good, please purified and rerun the proton and carbon.

A2.     Thank you very much for your comments.  The compound 4 is raw sample because of non-purified amine.  The product of the amine was filtered to remove Pd/C and used for the nest step without further purification.

Q3.     The compound 5 integral is not accurate with your paper, page 4, line 133.

Q3.     Thank you very much for your comments.  The cite position of number of protons at 6.07 ppm was arranged.